# Extensive Benchmarking of DFT+*U* Calculations for Predicting Band Gaps

**Nicole E. Kirchner-Hall** [1,*], **Wayne Zhao** [1], **Yihuang Xiong** [1], **Iurii Timrov** [2] **and Ismaila Dabo** [1]

[1] Department of Materials Science and Engineering and the Materials Research Institute, The Pennsylvania State University, University Park, PA 16802, USA; wuz75@psu.edu (W.Z.); yyx5048@psu.edu (Y.X.); dabo@psu.edu (I.D.)

[2] Theory and Simulation of Materials (THEOS) and National Centre for Computational Design and Discovery of Novel Materials (MARVEL), École Polytechnique Fédérale de Lausanne, CH-1015 Lausanne, Switzerland; iurii.timrov@epfl.ch (I.T.)

[*] Correspondence: nek5091@psu.edu

**Abstract:** Accurate computational predictions of band gaps are of practical importance to the modeling and development of semiconductor technologies, such as (opto)electronic devices and photoelectrochemical cells. Among available electronic-structure methods, density-functional theory (DFT) with the Hubbard *U* correction (DFT+*U*) applied to band edge states is a computationally tractable approach to improve the accuracy of band gap predictions beyond that of DFT calculations based on (semi)local functionals. At variance with DFT approximations, which are not intended to describe optical band gaps and other excited-state properties, DFT+*U* can be interpreted as an approximate spectral-potential method when *U* is determined by imposing the piecewise linearity of the total energy with respect to electronic occupations in the Hubbard manifold (thus removing self-interaction errors in this subspace), thereby providing a (heuristic) justification for using DFT+*U* to predict band gaps. However, it is still frequent in the literature to determine the Hubbard *U* parameters semiempirically by tuning their values to reproduce experimental band gaps, which ultimately alters the description of other total-energy characteristics. Here, we present an extensive assessment of DFT+*U* band gaps computed using self-consistent *ab initio U* parameters obtained from density-functional perturbation theory to impose the aforementioned piecewise linearity of the total energy. The study is carried out on 20 compounds containing transition-metal or *p*-block (group III-IV) elements, including oxides, nitrides, sulfides, oxynitrides, and oxysulfides. By comparing DFT+*U* results obtained using nonorthogonalized and orthogonalized atomic orbitals as Hubbard projectors, we find that the predicted band gaps are extremely sensitive to the type of projector functions and that the orthogonalized projectors give the most accurate band gaps, in satisfactory agreement with experimental data. This work demonstrates that DFT+*U* may serve as a useful method for high-throughput workflows that require reliable band gap predictions at moderate computational cost.

**Keywords:** DFT+*U*; Hubbard correction; Löwdin orthogonalization; band gap; self-interaction

## 1. Introduction

Density-functional theory (DFT) [1,2] with approximate exchange-correlation (xc) functionals—e.g., local-density approximation (LDA) or generalized-gradient approximation (GGA)—has been remarkably successful in predicting ground-state properties of a large variety of systems, such as crystal structure and thermodynamic stability. However, these DFT calculations have known limitations, including the underestimation of the band gap (on the order of ∼40% in semiconductors and insulators [3]) due to self-interaction errors (SIE) inherent to approximate xc functionals [4,5]. Various xc functionals and methods were developed to alleviate this problem. One practical solution is to use

hybrid functionals, which employ a fraction of the exact (Fock) exchange energy to reduce SIE [6–11]. The drawback of this method is its very high computational cost (which prevents its wider application to high-throughput computational studies [12]), and difficulties in determining the magnitude of the screening parameter for solids (that controls the amount of the exact exchange), although there has been considerable progress in this direction in recent years (see, e.g., Refs. [13–18]). In parallel, a range of computationally efficient meta-GGA functionals have emerged [19], including the strongly constrained and appropriately normed semilocal density functional (SCAN) [20]. The SCAN functional uses a xc potential which depends on Kohn-Sham (KS) wavefunctions via a kinetic energy density; however, SCAN still contains significant SIE (especially when applied to transition-metal compounds [21,22]) and exhibits some potential limitations in describing magnetic systems [23,24]. An alternative approach is provided by Koopmans-compliant (spectral) functionals [25]. The main idea of these is to enforce the piecewise linearity of the total energy with respect to all orbitals occupations; this class of functionals has proven to be effective in improving band gaps in finite and extended systems [26–29]; however, this method is not straightforward to implement in practice, though recent progress in this direction has been achieved [30]. A very popular extension of DFT is the DFT+$U$ approach [31–33], in which the Hubbard $U$ correction acts selectively on a subset of states in the system (typically, of $d$ or $f$ character) by imposing the piecewise linearity in the energy functional as a function of the occupations of this subset [34]. DFT+$U$ is only marginally more expensive than DFT within LDA or GGA, while significantly improving various properties of materials such as transition-metal compounds. An extension of DFT+$U$ to take into account inter-site Hubbard $V$ interactions (DFT+$U$+$V$ [35]) was introduced, showing success in describing materials with strong inter-site electronic hybridizations (i.e., covalent interactions) [35–41]. Finally, there are various methods beyond DFT, including many-body perturbation theory (MBPT) [42] (e.g., GW approximation [43–46]) and dynamical mean field theory (DMFT) [47–51], which are widely used for predicting optical properties of (strongly) correlated systems.

In contrast to MBPT and DMFT, the scope of DFT and DFT+$U$ is limited to ground-state properties, including electronic band gaps but excluding optical band gaps [28,52]. The optical band gap of a material is the minimal energy required for absorbing an incident photon (corresponding to a neutral excitation), whereas the electronic band gap is the difference between the occupied and unoccupied states (corresponding to charged excitations) [52]. In most inorganic semiconductors, the electronic band gap and the optical band gap are approximately equal, but for organic semiconductors, the difference between these two energies (the exciton binding energy) may be substantial. It is thus expected that DFT+$U$ would be more accurate at predicting optical band gaps for inorganic semiconductors than for organic semiconductors [53,54]. To improve the precision of the computed electronic band gaps, DFT+$U$ incorporates a corrective $U$ term designed to restore piecewise linearity of the total energy with respect to the occupations of the orbitals within the Hubbard manifold, which acts to approximate derivative discontinuities [28,35]. Often the $U$ parameter is chosen semiempirically by fitting it to reproduce experimental band gaps [12,55,56], experimental oxidation enthalpies [57,58], or other properties. However, fitting $U$ to reproduce a subset of experimental data is not a predictive approach, especially in cases when no prior data is available. If instead the Hubbard $U$ parameter is determined using *ab initio* methods, the predicted band gap might not exactly match the experimental gap, but this approach typically improves band gap predictions drastically (over semilocal DFT) and provides a more accurate, overall description of the material [12].

In practical terms, the Hubbard $U$ parameter can be calculated using *ab initio* methods, such as constrained DFT (cDFT) [34,59–67], constrained random phase approximation (cRPA) [68–74], and Hartree-Fock-based methods [75–78]. In particular, a linear-response formulation of cDFT was introduced in Ref. [34]: it is mainly based on the fact that the Hubbard corrections are meant to remove SIE from approximate energy functionals that manifests itself through a curvature of the total energy as a function of atomic occupations.

Recently, this formulation was recast in the framework of density-functional perturbation theory (DFPT) [38,79], allowing the replacement of computationally expensive supercells by primitive cells at the cost of having multiple monochromatic perturbations instead of just one (localized). In all these *ab initio* methods for computing $U$, key is the choice of the Hubbard projector functions. There are different types of Hubbard projectors [80], including nonorthogonalized atomic orbitals [34,81–83], orthogonalized atomic orbitals [36,37,39,84], nonorthogonalized Wannier functions [85], orthogonalized Wannier functions [86], linearized augmented plane-wave approaches [87], and projector-augmented-wave projector functions [88,89]. It is well known that the values of the computed Hubbard $U$ parameters strongly depend on the type of Hubbard projectors used [79,84]. Moreover, the final properties of interest (e.g., band gaps, oxidation reaction energy, among others) obtained using DFT+$U$ and different projectors are not always consistent [90]. Therefore, specific attention must be given to the choice of Hubbard projectors when computing band gaps using DFT+$U$.

Historically, DFT+$U$ was introduced to accurately describe the $d$- and $f$-type electrons of transition-metal (TM) and rare-earth compounds, as it was thought that the $U$ parameter would correct these strongly correlated, localized states (i.e., $d$- and $f$-type states). It was later discovered that the Hubbard $U$ parameter corrects for SIE, which is present in not only $d$ and $f$, but also $s$ and $p$ states [91,92]. While light elements (e.g., O, N, S) contain only $s$ and $p$ states and thus do not have as large SIE as TM, the $U$ correction has been increasingly applied to light elements in recent years to improve the computational prediction of properties, such as the band gap [78,93,94]. Additionally, in the literature, $U$ has been applied to $d^{10}$ elements from the $p$-block (specifically group III-IV) of the periodic table, such as $Ga^{2+}$ [35]. While elements, like $Ga^{2+}$, are not TM, they have electronic configurations that match that of a $d^{10}$ TM under certain ionizations. Applying $U$ to light elements and/or $d^{10}$ group III-IV elements (e.g., Ge, In, Sn, Pb) is still in debate in the literature [35,78,93,94]. This question will be addressed in this work using DFPT [79], and in particular we will explore the sensitivity of band gaps to the application of the Hubbard $U$ correction to $p$ states of light elements.

In this paper we present a benchmark of DFT+$U$ using *ab initio* Hubbard $U$ [79] for predicting band gaps in 20 compounds containing transition-metal or $p$-block (group III-IV) elements, including oxides, nitrides, sulfides, oxynitrides, and oxysulfides. This is done using two types of Hubbard projector functions, namely, nonorthogonalized atomic orbitals (that we will refer to as *atomic* orbitals, for simplicity) which are provided with pseudopotentials and which are orthonormal within each atom, and orthogonalized atomic orbitals which are obtained by orthogonalizing the atomic orbitals from different sites using the Löwdin method (that we will call *ortho-atomic* orbitals) [95]. By inspecting the dominant character of electronic states at the top of the valence bands and at the bottom of the conduction bands we select the most relevant states and apply the Hubbard correction to them and analyze the subsequent changes in the band gaps. Our study reveals that the DFT+$U$ band gaps are very sensitive to the type of Hubbard projectors, and that the most accurate results are obtained using *ortho-atomic* ones. Moreover, this work shows that DFT+$U$ with Hubbard parameters determined nonempirically can be used to predict reliable band gaps at lower computational cost than that of hybrid functionals (such as HSE06 [96,97]) that are popular for predicting band gap values. This is important for high-throughput studies that require efficient and reliable estimates of the band gaps at moderate computational cost for various technologically relevant materials such as photocatalysts [98,99], solar cells [100,101], and transparent conductors [102].

The remainder of the paper is organized as follows. Section 2 briefly recalls the formulation of the DFT+$U$ approach and how Hubbard $U$ is computed using DFPT, and describes which materials are studied in this work. Section 3 contains the technical details of our calculations. In Section 4 we present the projected density of states for all materials of this work, the Hubbard parameter values computed used *ab initio* methods, and the comparison of the computed band gaps with those measured experimentally. Finally,

Section 5 presents our concluding remarks. Appendix A reports our convergence tests when computing $U$, and Appendix B contains a comparison of the computational costs of HSE06 and linear-response $U$ calculations.

## 2. Methods and Materials

### 2.1. DFT+U

In this section, we briefly review DFT+$U$ in the simplified, rotationally invariant formulation [33] which is used in this work. For the sake of simplicity, the formalism is presented in the framework of norm-conserving pseudopotentials in the collinear spin-polarized case. The generalization to the ultrasoft pseudopotentials and the projector augmented wave method are described in Ref. [38]. In this section we use Hartree atomic units.

DFT+$U$ is based on an additive correction to the approximate DFT energy functional [33]:

$$E_{\text{DFT}+U} = E_{\text{DFT}} + E_U, \tag{1}$$

where $E_{\text{DFT}}$ is the approximate DFT energy (constructed, e.g., within the spin-polarized LDA or GGA), while $E_U$ contains the additional Hubbard term:

$$E_U = \frac{1}{2} \sum_{I\sigma m_1 m_2} U^I \left( \delta_{m_1 m_2} - n_{m_1 m_2}^{I\sigma} \right) n_{m_2 m_1}^{I\sigma}, \tag{2}$$

where $I$ is the atomic site index, $m_1$ and $m_2$ are the magnetic quantum numbers associated with a specific angular momentum, and $U^I$ are the effective on-site Hubbard parameters. The atomic occupation matrices $n_{m_1 m_2}^{I\sigma}$ are based on a projection of the lattice-periodic parts of KS wavefunctions, $u_{v,\mathbf{k}}^{\sigma}(\mathbf{r})$, on the Hubbard manifold:

$$n_{m_1 m_2}^{I\sigma} = \frac{1}{N_{\mathbf{k}}} \sum_{\mathbf{k}}^{N_{\mathbf{k}}} \sum_{v} f_{v,\mathbf{k}}^{\sigma} \langle u_{v,\mathbf{k}}^{\sigma} | \hat{P}_{m_2 m_1 \mathbf{k}}^{I} | u_{v,\mathbf{k}}^{\sigma} \rangle, \tag{3}$$

where $v$ and $\sigma$ represent, respectively, the band and spin labels of the KS wavefunctions, $\mathbf{k}$ indicate points in the first Brillouin zone (BZ) and $N_{\mathbf{k}}$ is the number of $\mathbf{k}$-points, $f_{v,\mathbf{k}}^{\sigma}$ are the occupations of the KS states, and $\hat{P}_{m_2 m_1 \mathbf{k}}^{I}$ is the projector on the Hubbard manifold:

$$\hat{P}_{m_2 m_1 \mathbf{k}}^{I} = |\varphi_{m_2 \mathbf{k}}^{I}\rangle \langle \varphi_{m_1 \mathbf{k}}^{I}|, \tag{4}$$

where $\varphi_{m_1 \mathbf{k}}^{I}(\mathbf{r})$ is the Bloch sum computed using $\varphi_{m_1}^{I}(\mathbf{r} - \mathbf{R}_I)$ which are the localized orbitals centered on the $I$th atom at the position $\mathbf{R}_I$. The Hubbard manifold can be constructed using different types of projector functions, as was discussed in Section 1; here we consider two types of localized functions, *atomic* and *ortho-atomic*.

In DFT+$U$, the Hubbard parameters are not known *a priori*, and they are often adjusted semiempirically, by matching the value of properties of interest, which is fairly arbitrary. In this work we instead determine Hubbard parameters from a piecewise linearity condition implemented through linear-response theory [34], based on DFPT [38,79]. Within this framework the Hubbard parameters are the elements of an effective interaction matrix computed as the difference between bare and screened inverse susceptibilities [34]:

$$U^I = \left( \chi_0^{-1} - \chi^{-1} \right)_{II}, \tag{5}$$

where $\chi_0$ and $\chi$ are the susceptibilities which measure the response of atomic occupations to shifts in the potential acting on individual Hubbard manifolds. In particular, $\chi$ is defined as

$$\chi_{IJ} = \sum_{\sigma m} \frac{dn_{mm}^{I\sigma}}{d\alpha^J}, \tag{6}$$

where $\alpha^J$ is the strength of the perturbation on the $J$th site. While $\chi$ is evaluated at self-consistency of the linear-response KS calculation, $\chi_0$ (which has a similar definition) is computed before the self-consistent re-adjustment of the Hartree and exchange-correlation potentials [34]. The main goal of the DFPT implementation is to recast the response to such isolated perturbations in supercells as a sum over a regular grid of $N_\mathbf{q}$ $\mathbf{q}$-points in the Brillouin zone [79]

$$\frac{dn_{mm'}^{I\sigma}}{d\alpha^J} = \frac{1}{N_\mathbf{q}} \sum_\mathbf{q}^{N_\mathbf{q}} e^{i\mathbf{q}\cdot(\mathbf{R}_l - \mathbf{R}_{l'})} \Delta_\mathbf{q}^{s'} n_{mm'}^{s\sigma} . \tag{7}$$

It is important to note that the monochromatic responses in Equation (7) are calculated in the primitive unit cell, thus avoiding the use of the computationally expensive supercells. In Equation (7), the atomic indices have been replaced by atomic ($s$ and $s'$) and unit cell ($l$ and $l'$) labels [$I \equiv (l, s)$ and $J \equiv (l', s')$]; $\mathbf{R}_l$ and $\mathbf{R}_{l'}$ are the Bravais lattice vectors, and the grid of $\mathbf{q}$-points is chosen fine enough to make the resulting atomic perturbations effectively decoupled from their periodic replicas [38,79]. Since $\Delta_\mathbf{q}^{s'} n_{mm'}^{s\sigma}$ are the lattice-periodic responses of atomic occupations to a monochromatic perturbation of wavevector $\mathbf{q}$, they can be obtained by solving the DFPT equations independently for every $\mathbf{q}$, which allows us to parallelize and speed up the calculations of the Hubbard parameters.

### 2.2. Materials

To cover a sufficiently representative set of materials, we have investigated binary oxides, binary nitrides, binary sulfides, oxynitrides, and oxysulfides. Metals in these compounds were limited to $d^0$ ($Sc^{3+}$, $Y^{3+}$, $Ti^{4+}$, $Zr^{4+}$, $Hf^{4+}$, $V^{5+}$, $Nb^{5+}$, $Ta^{5+}$, $Mo^{6+}$, $W^{6+}$) and $d^{10}$ ($In^{3+}$, $Ge^{4+}$, $Sn^{4+}$, $Pb^{4+}$) cations to tailor the study towards photocatalytic materials [99]. Using the data from the Materials Project [103], we created combinations of O, N, S, and these $d^0$ and $d^{10}$ cations. This resulted in an extensive list of materials which was paired down using criteria of unit cell size and experimental band gap: the number of atoms per unit cell was limited to < 20 and an experimental band gap had to be available in the literature for comparison. Table 1 shows the resulting list, containing 8 oxides, 2 nitrides, 6 sulfides, 3 oxynitrides, and 1 oxysulfide. These materials are all $d^0$ or $d^{10}$ closed-shell systems due to the oxidation states of the TM and group III-IV elements, as listed above.

**Table 1.** Chemical formula and space group for each of the materials in this benchmark study.

|   | Formula | Space Group |
|---|---|---|
| 1 | $TiO_2$ | $P4_2/mnm$ |
| 2 | $ZrO_2$ | $P2_1/c$ |
| 3 | $HfO_2$ | $P2_1/c$ |
| 4 | $V_2O_5$ | $Pmmn$ |
| 5 | $Ta_2O_5$ | $C2/c$ |
| 6 | $WO_3$ | $Pm\bar{3}m$ |
| 7 | $SnO_2$ | $P4_2/mnm$ |
| 8 | $PbO_2$ | $P4_2/mnm$ |
| 9 | $InN$ | $P6_3mc$ |
| 10 | $Sn_3N_4$ | $Fd\bar{3}m$ |
| 11 | $TiS_2$ | $P\bar{3}m1$ |
| 12 | $ZrS_2$ | $P\bar{3}m1$ |
| 13 | $HfS_2$ | $P\bar{3}m1$ |
| 14 | $MoS_3$ | $P2_1/m$ |
| 15 | $Sc_2S_3$ | $Fddd$ |
| 16 | $SnS_2$ | $P\bar{3}m1$ |
| 17 | $NbNO$ | $P2_1/c$ |
| 18 | $TaNO$ | $P2_1/c$ |
| 19 | $Ge_2N_2O$ | $Cmc2_1$ |
| 20 | $Y_2SO_2$ | $P\bar{3}m1$ |

## 3. Technical details

All calculations were performed using the plane-wave (PW) pseudopotential method as implemented in the QUANTUM ESPRESSO distribution [104–106]. We used GGA for the xc functional constructed with the PBE prescription [107], and we selected the ultrasoft pseudopotentials from the GBRV library [108,109]. KS wavefunctions and charge density were expanded in plane waves up to a kinetic-energy cutoff of 60 and 480 Ry, respectively. Electronic ground states were computed by sampling the BZ with uniform Γ-centered **k**-point meshes with a spacing of 0.04 Å$^{-1}$.

DFT+$U$ calculations were performed using the simplified rotationally invariant formulation [33]. As projectors for the Hubbard manifold, we used *atomic* and *ortho-atomic* orbitals [84]; for the latter we employed the Löwdin orthogonalization method [95]. Hubbard $U$ parameters were computed using the linear-response approach [34] as implemented using DFPT [79] (see Section 2.1). We have used the self-consistent procedure for the calculation of $U$ as described in detail in Ref. [38], which consists of cyclic calculations containing structural optimizations and recalculations of Hubbard parameters for each new geometry. We have performed convergence tests with respect to the **q**-point sampling for 3 materials (TiO$_2$, NbNO, Y$_2$SO$_2$); the convergence criteria was $\Delta U < 0.1$ eV. The details can be found in the Appendix A. After selecting the **q**-point sampling, hybrid functional (HSE06) calculations were performed for two representative materials (TiO$_2$ and NbNO) to compare the computational costs of HSE06 and $U$ calculations. The results of this analysis are detailed in Appendix B.

All materials were tested for possible finite magnetization by performing collinear spin-polarized calculations within standard DFT. In order to allow for fractional occupations we have used the Marzari-Vanderbilt smearing [110] with a broadening parameter of $5 \times 10^{-3}$ Ry. In this case we also sampled the BZ using the uniform Γ-centered **k**-point meshes with a spacing of 0.04 Å$^{-1}$. Materials were only tested for ferromagnetic ordering; thus, this study is not exhaustive and more testing would be needed to check for other types of possible magnetic orderings. Based on our computational testing for ferromagnetic ordering, all of the materials in this study showed zero magnetization. Therefore, all materials were modeled as nonmagnetic, which could lead to errors in predicted band gap values for materials that do show magnetic ordering other than ferromagnetic. In fact, some of the materials are known to be antiferromagnetic (V$_2$O$_5$) or ferromagnetic (SnO$_2$, SnS$_2$) in their defective forms [111–113], but the primary focus of this work is to investigate the pristine bulk phase.

We have computed the projected density of states (PDOS) using both DFT and DFT+$U$ with the Gaussian smearing and a broadening parameter of $5 \times 10^{-3}$ Ry. The PDOS was calculated by summing up states from all equivalent atoms in the unit cell.

## 4. Results and Discussion

### 4.1. Determination of the States Requiring Hubbard Corrections

One can assess the necessity of applying the Hubbard $U$ correction to certain states of a given element by examining the PDOS of the material. In particular, the PDOS plots show which specific states have the highest density of states (DOS) near the valence band maximum (VBM) and conduction band minimum (CBM) in semiconductors and insulators. By applying $U$ corrections to elements with states at the VBM and CBM, the value of the band gap is expected to change with respect to standard DFT predictions. In contrast, applying $U$ values to elements with low DOS or lacking states near band edges is not expected to significantly affect the predicted band gap value. Using PDOS, we can also assess whether a given material is a band insulator, a Mott-Hubbard insulator, or a charge-transfer insulator [114]. In Mott-Hubbard insulators, the VBM and CBM are of the same kind (e.g., of $d$ character), in charge-transfer insulators instead the VBM and CBM are of different kinds (e.g., the VBM is mainly of the $p$ character and the CBM is of the $d$ character), while in band insulators the band gap is not determined by strongly localized electrons of

*d* or *f* character. Therefore, by determining the dominant character of the VBM and CBM in the PDOS, it is possible to identify the insulating type of a given material.

Thus, prior to DFT+*U* calculations, PDOS calculations were performed in order to determine the dominant character of states at the VBM and CBM, and thus to know which states might require the Hubbard *U* correction. The PDOS for all materials studied here are shown in Figure 1. It can be seen that all 14 TM-containing materials can be classified as charge-transfer insulators, while all of the remaining materials that contain group III-IV elements (except InN that comes out to be metallic at the semilocal DFT level of theory) are band insulators. Also we note that in Figure 1 those cases that have multiple lines of the same color indicate the PDOS for non-equivalent atoms (e.g., multiple red lines for $V_2O_5$ indicate 2*p* states of non-equivalent oxygen atoms).

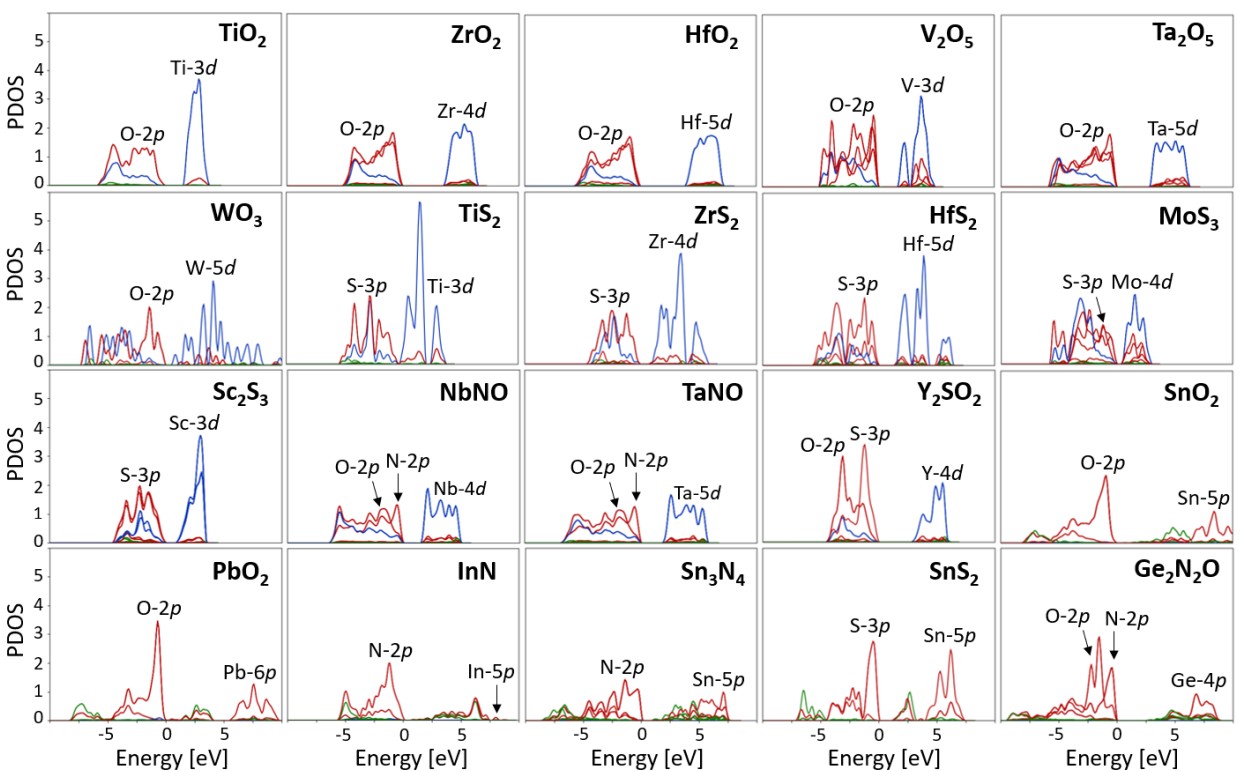

**Figure 1.** PDOS (in states/eV/cell) for all materials studied in this work obtained from standard DFT calculations. PDOS are color-coordinated such that *s* states are green, *p* states are red, and *d* states are blue. Note that cases where there are multiple lines of the same color indicate non-equivalent atoms. The zero of energy corresponds to the top of valence bands (except for InN which comes out to be metallic at the semilocal DFT level, and hence the zero of energy corresponds in this case to the Fermi level).

All the materials of this study are oxides, nitrides, and sulfides containing $d^0/d^{10}$ cations. The PDOS of $d^0$ TM-containing materials ($TiO_2$, $ZrO_2$, $HfO_2$, $V_2O_5$, $Ta_2O_5$, $WO_3$, $TiS_2$, $ZrS_2$, $HfS_2$, $MoS_3$, $Sc_2S_3$, NbNO, TaNO, $Y_2SO_2$) in Figure 1 show dominant *d* states from the TM $d^0$ cations at the CBM and dominant *p* states from the light elements (O, N, S) at the VBM. Hence, one might expect that the application of the Hubbard *U* correction to these states would allow us to obtain the most accurate prediction of the band gap at the DFT+*U* level of theory. The PDOS of $d^{10}$ *p*-block (group III-IV) containing materials ($SnO_2$, $PbO_2$, InN, $Sn_3N_4$, $SnS_2$, $Ge_2N_2O$) show dominant *p* states from the light elements (O, N, S) at the VBM and a mixture of *p* and *s* states from the group III-IV elements at the CBM, indicating that the *U* correction can be applied to the *p* states of the light and group III-IV elements to obtain the most accurate prediction of the band gap [115]. In addition, it might be useful to investigate in these materials whether the application of the *U* correction only to the *d* states of the TM elements in $d^0$ TM-containing materials or only to the *p* states of

the light elements (i.e., at the VBM) in $d^{10}$ $p$-block containing materials would be sufficient to improve band gaps with respect to standard DFT. Moreover, in this latter case we will show that the application of the $U$ correction to the $p$ states at the CBM turns out to be not so important, since this leads only to minor changes in the band gaps within the semilocal DFT approximation. A similar test for $d^0$ TM-containing materials will not be performed because the $d$ states of TM elements appear not only at the CBM but also in the valence region, and show significant hybridization with the $p$ states of light elements, which means that the application of $U$ to the $d$ states of TM elements is expected to be important. Finally, we note that InN is metallic at the generalized-gradient level (in Figure 1 the DOS above the Fermi level is very small and is not well visible). The values of Hubbard parameters computed for the VBM and CBM states discussed above are reported below.

### 4.2. Hubbard U Parameters from First Principles

In this section we present the Hubbard parameters for all systems studied here, computed using the DFPT approach that was briefly discussed in Section 2.1. Based on the PDOS computed at the semilocal DFT level (see Section 4.1), we calculated $U$ for those states that appear at the VBM and CBM, and the results are shown in Table 2. More specifically, for the $d^0$ TM-containing materials we computed the Hubbard parameters for the partially occupied $d$ states of TM elements (Sc, Ti, V, Y, Zr, Nb, Mo, Hf, Ta, W) that appear at the CBM and also for the $p$ states of light elements (O, N, S) that appear at the VBM. In the case of $d^{10}$ $p$-block (group III-IV) containing materials we computed the Hubbard $U$ parameters for the $p$ states of light elements at the VBM and for the $p$ states of the group III-IV elements the CBM.

**Table 2.** Comparison of the Hubbard $U$ parameters (in eV) as computed using DFPT with *atomic* and *ortho-atomic* Hubbard projectors. Numbered atoms (e.g., O1, O2) indicate nonequivalent atoms which require individual $U$ values.

| Formula | Hubbard $U$ (Ortho-Atomic) [eV] | | | Hubbard $U$ (Atomic) [eV] | | | |
|---|---|---|---|---|---|---|---|
| TiO$_2$ | Ti(3d): 6.10, | O(2p): 8.23 | | Ti(3d): 3.81, | O(2p): 15.89 | | |
| ZrO$_2$ | Zr(4d): 2.72, | O1(2p): 9.20, | O2(2p): 8.79 | Zr(4d): 1.07, | O1(2p): 33.67, | O2(2p): 27.20 | |
| HfO$_2$ | Hf(5d): 2.74, | O1(2p): 9.58, | O2(2p): 9.41 | Hf(5d): 1.14, | O1(2p): 47.79, | O2(2p): 38.18 | |
| V$_2$O$_5$ | V(3d): 5.37, O3(2p): 7.06 | O1(2p): 7.42, | O2(2p): 7.65, | V(3d): 3.70, O3(2p): 11.12 | O1(2p): 12.96, | O2(2p): 10.09, | |
| Ta$_2$O$_5$ | Ta1(5d): 3.06, O2(2p): 8.36, | Ta2(5d): 3.07, O3(2p): 8.20 | O1(2p): 8.99, | Ta1(5d): 1.54, O2(2p): 30.10, | Ta2(5d): 1.55, O3(2p): 19.03, | O1(2p): 30.11, | O4(2p): 17.79 |
| WO$_3$ | W(5d): 3.99, | O(2p): 8.27 | | W(5d): 2.80, | O(2p): 15.18 | | |
| TiS$_2$ | Ti(3d): 5.61, | S(3p): 3.85 | | Ti(3d): 4.45, | S(3p): 6.39 | | |
| ZrS$_2$ | Zr(4d): 2.61, | S(3p): 3.96 | | Zr(4d): 1.62, | S(3p): 8.36 | | |
| HfS$_2$ | Hf(5d): 2.61, | S(3p): 3.75 | | Hf(5d): 1.81, | S(3p): 9.65 | | |
| MoS$_3$ | Mo(4d): 3.28, S3(3p): 4.62 | S1(3p): 3.31, | S2(3p): 4.61, | Mo(4d): 3.87, S3(3p): 5.53 | S1(3p): 6.08, | S2(3p): 5.62, | |
| Sc$_2$S$_3$ | Sc1(3d): 3.28, S2(3p): 3.88 | Sc2(3d): 3.37, | S1(3p): 3.83, | Sc1(3d): 1.69, S2(3p): 9.15, | Sc2(3d): 1.77, S3(3p): 9.17 | S1(3p): 8.72, | |
| NbNO | Nb(4d): 3.48, | N(2p): 6.47, | O(2p): 8.51 | Nb(4d): 2.17, | N(2p): 9.85, | O(2p): 19.94 | |
| TaNO | Ta(5d): 3.10, | N(2p): 6.66, | O(2p): 8.87 | Ta(5d): 1.88, | N(2p): 11.51, | O(2p): 26.04 | |
| Y$_2$SO$_2$ | Y(4d): 1.98, | S(3p): 4.16, | O(2p): 9.17 | Y(4d): 0.59, | S(3p): 28.16, | O(2p): 47.87 | |
| SnO$_2$ | O(2p): 10.19, | Sn(5p): 1.26 | | O(2p): 95.85, | Sn(5p): 0.67 | | |
| PbO$_2$ | O(2p): 9.58, | Pb(6p): 1.20 | | O(2p): 39.60, | Pb(6p): 0.58 | | |
| InN | N(2p): 6.52, | In(5p): 1.17 | | N(2p): 28.91, | In(5p): 0.62 | | |
| Sn$_3$N$_4$ | N(2p): 6.56, | Sn1(5p): 1.43, | Sn2(5p): 1.23 | N(2p): 24.14, | Sn1(5p): 1.01, | Sn2(5p): 0.81 | |
| SnS$_2$ | S(3p): 3.88, | Sn(5p): 1.37 | | S(3p): 7.66, | Sn(5p): 1.16 | | |
| Ge$_2$N$_2$O | O(2p): 10.35, | N(2p): 6.62, | Ge(4p): 1.18 | O(2p): 103.08, | N(2p): 31.57, | Ge(4p): 0.72 | |

First of all it is important to discuss the dependence of the computed values of the $U$ parameters on the type of Hubbard projectors that are used. In this work we consider two types of projector functions, namely *atomic* and *ortho-atomic*. The difference between the two is the inter-site orthogonalization of atomic orbitals that is present in the latter

and is absent in the former. As can be seen in Table 2, the values of Hubbard parameters depend strongly on the type of projector functions. In particular, we can see that $U$ for the $d$ states of TM elements is larger when *ortho-atomic* projectors are used, while $U$ for the $p$ states of light elements are much smaller with respect to the case when *atomic* projectors are employed. Notably, $U$ for the $p$ states of light elements in some materials are extremely large, and in fact this finding is not surprising and is well known in the literature [116]: linear-response theory for computing Hubbard parameters according to Ref. [34] is not suited for application to fully occupied states (closed-shell systems), which was observed when using *atomic* Hubbard projectors. Indeed, it can be seen from Table 2 that the values of $U$ for the $p$ states of light elements (O, N, S) are unphysically large when computed using *atomic* projectors. As will be seen in Section 4.3, this leads to dramatic worsening of the band gaps when computed using DFT+$U$ with these values of $U$ and *atomic* projectors. However, interestingly we find that when *ortho-atomic* Hubbard projectors are used, the values of Hubbard $U$ for the $p$ states of light elements are much smaller and they fall in a physically meaningful range (though still quite large). As will be seen in the following section, the band gaps computed using DFT+$U$ and using these $U$ values with the *ortho-atomic* projectors are in good overall agreement with the experimental band gaps. Therefore, this observation highlights the crucial role played by the Hubbard projectors [90] when computing $U$ for predicting various properties of materials, and in particular band gaps.

Next, in the case of $d^0$ TM-containing materials, the $U$ values computed for the partially occupied $d$ states were applied to either $3d$, $4d$, or $5d$ states depending on the TM period in the periodic table. In general, the smaller the principal quantum number the larger the localization of the $d$ orbitals (because they are closer to the nucleus); thus, the value of $U$ is expected to be larger for $3d$ states than for $4d$ and $5d$ states [117]. Indeed, we see this trend for the $U$ values of the TM $d^0$ elements in Table 2. In the case of the $d^{10}$ $p$-block containing materials with *ortho-atomic* Hubbard projectors we find that the $U$ values of the $p$ states of the group III-IV elements are very small ($< 1.5$ eV), which is not surprising since these states are almost fully empty. It will be shown in Section 4.3 that the application of the $U$ correction to the $p$ states of the group III-IV elements has only a minor effect on the computed band gaps.

### 4.3. Band Gaps from DFT+U Calculations

In this section we present our findings for the band gaps of all 20 materials computed using standard DFT and using DFT+$U$ with *ab initio* values of Hubbard $U$ (see Section 4.2) together with *atomic* and *ortho-atomic* Hubbard projectors. The resulting predicted band gaps are plotted in Figure 2a, for the materials containing TM elements, and in Figure 2b, for the materials containing $p$-block (group III-IV) elements; this data is also summarized in Tables 3 and 4.

We use the following short-hand notations: (*i*) in the TM-containing materials we use "$U$-TM" for DFT+$U$ calculations with the $U$ correction applied only to the $d$ states of TM elements, and "$U$-All" for DFT+$U$ calculations with the $U$ correction applied both to the $d$ states of the TM elements and to the $p$ states of the light elements (O, N, S); (*ii*) in the materials containing the $p$-block (group III-IV) elements we use "$U$-Light" for DFT+$U$ calculations with the $U$ correction applied only to the $p$ states of the light elements, and "$U$-All" for DFT+$U$ calculations with the $U$ correction applied both to the $p$ states of the light and group III-IV elements (Ge, In, Sn, Pb).

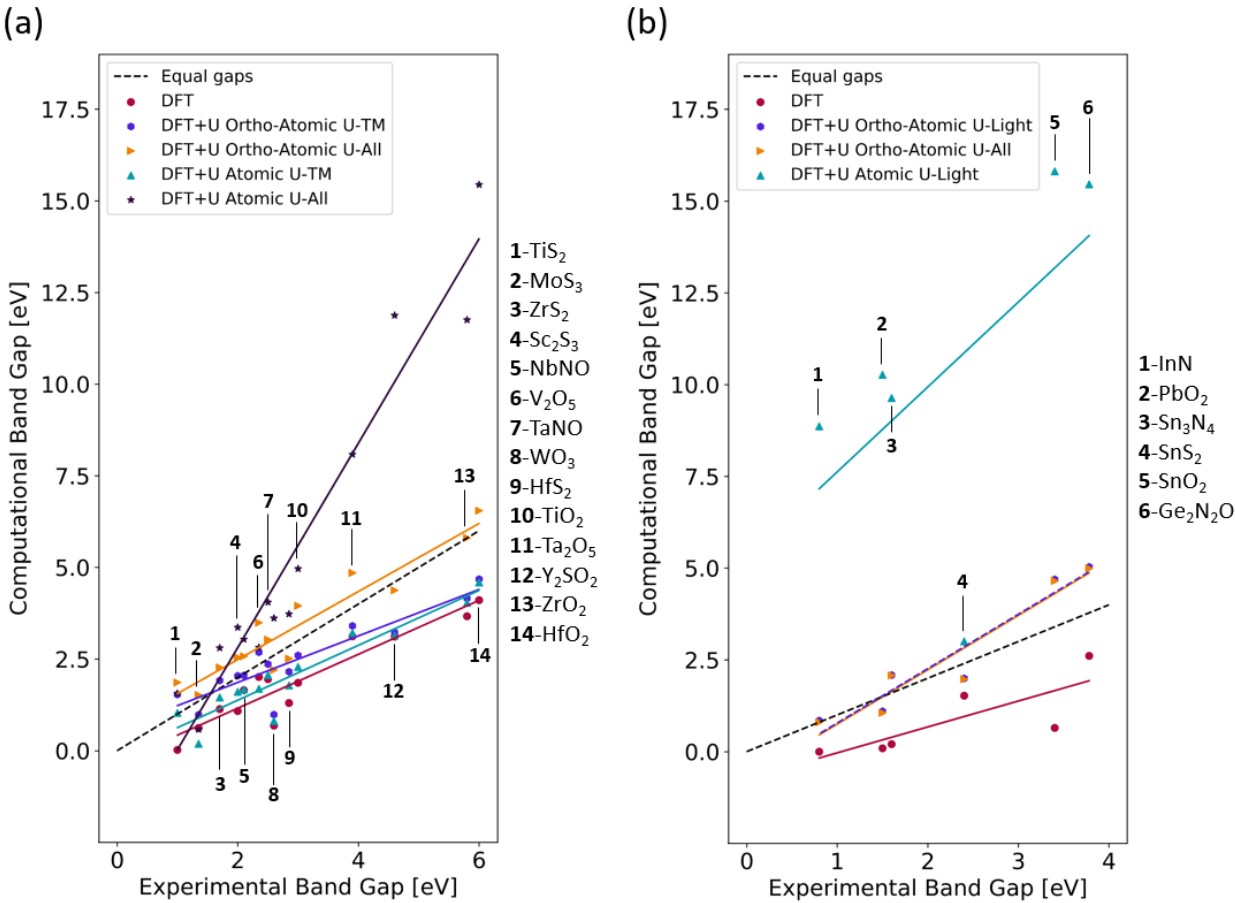

**Figure 2.** Comparison of the computationally predicted band gaps (from DFT and DFT+$U$) and the experimental band gaps for materials containing (**a**) TM elements, and (**b**) $p$-block (group III-IV) elements of the periodic table. The dashed black line indicates where the computational band gap equals the experimental band gap. Data was fit to first order polynomials so that data points from DFT and different DFT+$U$ calculations could be more easily compared using the resulting solid trendlines (note that all data points were included in these fits). DFT+$U$ calculations were performed using *atomic* and *ortho-atomic* projectors. On panel (**a**), the legend also indicates whether $U$ corrections are applied to only $d$ states of the TM elements ($U$-TM), or to the $d$ states of TM and $p$ states of light elements ($U$-All). On panel (**b**), the legend also indicates whether $U$ corrections are applied to $p$ states of light elements ($U$-Light), or to $p$ states of light and $p$-block (group III-IV) elements ($U$-All). Note that in panel (**b**), the "DFT+$U$ Ortho-Atomic $U$-Light" trendline (blue) is dashed so that the "DFT+$U$ Ortho-Atomic $U$-All" trendline (orange) can also be seen.

Upon inspection of the results for the TM-containing materials (Figure 2a) we can see large variations in the computed band gaps depending on the type of projector functions that are used (e.g., compare the slope of the lines obtained by fitting the data points). Overall, very good agreement with the experimental band gap values is obtained for DFT+$U$ calculations using *ortho-atomic* projectors with respective $U$ values. In contrast, DFT+$U$ calculations with the *atomic* projectors result in large, unrealistic values (>8 eV) for the band gaps in some materials ($Ta_2O_5$, $Y_2SO_2$, $ZrO_2$, and $HfO_2$) when Hubbard corrections are applied to TM and light elements (indicated by $U$-All) and underestimated band gaps when Hubbard corrections are applied to only TM (indicated by $U$-TM). In the case of *ortho-atomic* projectors, both $U$-TM and $U$-All results are in good agreement (on average) with the experimental band gaps, with $U$-All fitting line (mean absolute error of 0.55) being closer to the exact trend than the $U$-TM one (mean absolute error of 0.66). This latter finding can be further clarified by looking at Table 3: we can see that when the *ortho-atomic* projectors are used, both $U$-TM and $U$-All corrections perform equally well (which one is better depends on the material). It is worth highlighting the outliers from these trends: $V_2O_5$ shows comparable accuracy between band gaps computed using

*ortho-atomic* projectors with Hubbard corrections applied to only TM and standard DFT, and $TiS_2$ shows higher accuracy for the band gap computed with DFT+$U$ calculations using *atomic* projectors. For $V_2O_5$, it is important to remark that the antiferromagnetic ordering seen experimentally in $V_2O_5$ nanostructures may lead to inaccurate predictions of the band gap using DFT and DFT+$U$ since all materials in this study were modeled as nonmagnetic (see Section 3).

**Table 3.** Comparison of the band gaps (in eV) as computed using DFT and DFT+$U$ (using *ortho-atomic* and *atomic* projectors) and as measured in experiments for compounds containing TM elements, with the mean absolute error (MAE) given for each method. DFT+$U$ results obtained with *ortho-atomic* and *atomic* projectors are split into two columns each: (1) band gaps predicted with $U$ corrections applied only to the $d$ states of TM elements ($U$-TM), and (2) band gaps predicted with $U$ corrections applied to the $d$ states of TM and $p$ states of light elements ($U$-All). The computationally predicted band gaps with the closest values to the experimental band gaps are highlighted in bold.

| Formula | Expt. | DFT | DFT+$U$ (Ortho-Atomic) | | DFT+$U$ (Atomic) | |
| | | | $U$-TM | $U$-All | $U$-TM | $U$-All |
|---|---|---|---|---|---|---|
| $TiO_2$ | 3 [118] | 1.85 | **2.60** | 3.95 | 2.28 | 4.96 |
| $ZrO_2$ | 5.8 [119] | 3.67 | 4.16 | **5.81** | 4.04 | 11.75 |
| $HfO_2$ | 6 [120] | 4.11 | 4.68 | **6.55** | 4.59 | 15.44 |
| $V_2O_5$ | 2.35 [121] | **2.01** | 2.69 | 3.50 | 1.69 | 2.82 |
| $Ta_2O_5$ | 3.9 [122] | 3.12 | **3.41** | 4.85 | 3.22 | 8.08 |
| $WO_3$ | 2.6 [123] | 0.69 | 0.99 | **2.20** | 0.81 | 3.62 |
| $TiS_2$ | 1 [124] | 0.02 | 1.53 | 1.86 | **1.03** | 1.56 |
| $ZrS_2$ | 1.7 [125] | 1.13 | **1.92** | 2.27 | 1.45 | 2.81 |
| $HfS_2$ | 2.85 [126] | 1.30 | 2.16 | **2.51** | 1.78 | 3.73 |
| $MoS_3$ | 1.35 [127] | 0.62 | 0.98 | **1.52** | 0.19 | 0.58 |
| $Sc_2S_3$ | 2 [128] | 1.08 | **2.04** | 2.55 | 1.61 | 3.36 |
| NbNO | 2.1 [129] | 1.65 | **2.05** | 2.58 | 1.67 | 3.04 |
| TaNO | 2.5 [129] | 1.95 | **2.36** | 3.03 | 2.07 | 4.05 |
| $Y_2SO_2$ | 4.6 [130] | 3.11 | 3.23 | **4.38** | 3.14 | 11.88 |
| MAE | | 1.10 | 0.66 | 0.55 | 0.87 | 2.68 |

Materials containing $p$-block (group III-IV) elements show even more striking differences between the band gaps computed using *atomic* and *ortho-atomic* projectors with their corresponding values of $U$ parameters (see Figure 2b). Notably, it can be seen that the DFT+$U$ band gaps computed using *atomic* projectors are much larger than the experimental gaps, and, surprisingly, they are even worse than DFT band gaps. However, DFT+$U$ band gaps computed using *ortho-atomic* projectors with respective $U$ values are in good overall agreement with the experimental band gaps, and importantly these are better than the DFT band gaps. In addition we find that in the majority of cases $U$-All corrections give slightly better band gaps than $U$-Light, though these differences are very small. This latter finding suggests that the application of the $U$ correction to the $p$ states of the $p$-block (group III-IV) elements is not very important. Thus, $U$ values were not applied to the $p$ states of group III-IV elements for the case of DFT+$U$ band gaps computed using *atomic* projectors.

Therefore, from our analysis of the band gaps computed using DFT+$U$ for all of the 20 materials considered here, we can conclude that the Hubbard projectors play a fundamental role in the accuracy of band gap predictions. More specifically, depending on the type of projector functions, the Hubbard $U$ parameters vary dramatically (see Section 4.2) and as a consequence the DFT+$U$ band gaps also vary largely. Hence, the projector functions must be chosen with care prior to DFT+$U$ calculations. In particular, in our study we find that the *ortho-atomic* Hubbard projectors give band gaps that are in much better agreement with the experimental values, compared to DFT+$U$ with the *atomic* projectors. This result should be taken into account in future DFT+$U$ studies. In addition, it is important to mention that from Tables 3 and 4 we can see that still our best DFT+$U$ band gaps are not perfect and differ from the experimental ones by as much as almost ∼40% in some cases (e.g., $SnO_2$). This

finding suggests that DFT+$U$ with *ortho-atomic* projectors is a valuable tool for *improving* band gaps over standard DFT and thus can be useful for high-throughput screening of thousands of materials, but whenever one is interested in very accurate predictions of band gaps then it is important to resort to more advances and more accurate methods (see Section 1).

**Table 4.** Comparison of the band gaps (in eV) as computed using DFT and DFT+$U$ (using *ortho-atomic* and *atomic* projectors) and as measured in experiments for compounds containing group III-IV elements (Ge, In, Sn, Pb), with the mean absolute error (MAE) given for each method. DFT+$U$ calculations using *ortho-atomic* projectors are split into two columns: (1) band gaps predicted with $U$ corrections applied only to the $p$ states of light elements O, N, and S ($U$-Light), and (2) band gaps predicted with $U$ corrections applied to the $p$ states of group III-IV and light elements ($U$-All). For *atomic* calculations, band gaps were predicted with $U$ corrections applied only to the $p$ states of light elements ($U$-Light). The computationally predicted band gaps with the closest values to the experimental band gaps are highlighted in bold.

| Formula | Expt. | DFT | DFT+$U$ (Ortho-Atomic) | | DFT+$U$ (Atomic) |
| | | | $U$-Light | $U$-All | $U$-Light |
| --- | --- | --- | --- | --- | --- |
| $SnO_2$ | 3.4 [131] | 0.65 | 4.69 | **4.65** | 15.81 |
| $PbO_2$ | 1.5 [132] | 0.09 | **1.10** | 1.06 | 10.27 |
| InN | 0.8 [133] | 0 | 0.85 | **0.80** | 8.87 |
| $Sn_3N_4$ | 1.6 [134] | 0.21 | 2.09 | **2.08** | 9.63 |
| $SnS_2$ | 2.4 [135] | 1.53 | **2.00** | 1.98 | 3.00 |
| $Ge_2N_2O$ | 3.78 [136] | **2.61** | 5.03 | 4.99 | 15.46 |
| MAE | | 1.40 | 0.65 | 0.63 | 8.26 |

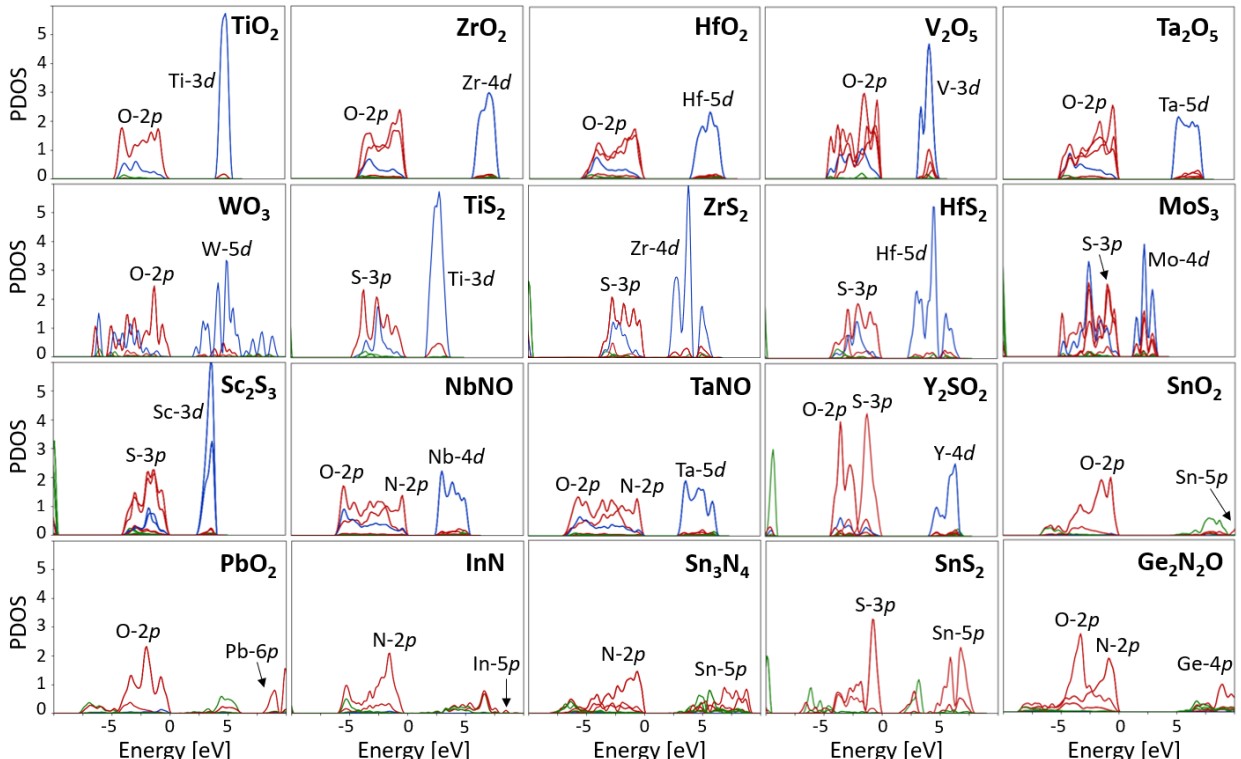

**Figure 3.** PDOS (in states/eV/cell) for all materials studied in this work obtained using DFT+$U$ with *ortho-atomic* projectors. The $U$ correction was applied both to the $d$ states of TM elements and to the $p$ states of the light and $p$-block (group III-IV) elements (i.e., $U$-All). PDOS are color-coordinated such that $s$ states are green, $p$ states are red, and $d$ states are blue. The zero of energy corresponds to the top of valence bands.

Finally, for the sake of completeness, we present the PDOS for all 20 materials using DFT+*U* with *ortho-atomic* projectors in Figure 3. The *U* correction was applied both to the *d* states of TM elements and to the *p* states of the light and *p*-block (group III-IV) elements (i.e., *U*-All). By comparing Figure 3 with Figure 1 we can see not only an increase in the value of the band gap thanks to the *U* correction but also "sharpening" of the PDOS due to the higher level of localization of states that are corrected. Although a comparative analysis of the PDOS on the basis of experimental spectroscopic data is beyond the scope of the present work, it would be an instructive topic for future studies.

## 5. Conclusions

We have presented a detailed investigation of the accuracy of DFT+*U* calculations for 20 materials containing transition-metal or *p*-block (group III-IV) elements, including oxides, nitrides, sulfides, oxynitrides, and oxysulfides. Hubbard *U* parameters were computed from first principles using density-functional perturbation theory [38,79], thus avoiding any fitting or tuning parameters. We have found that the accuracy of DFT+*U* band gaps depends strongly on the type of Hubbard projector functions used for applying the *U* corrections. More specifically, by comparing DFT+*U* results obtained using nonorthogonalized and orthogonalized atomic orbitals as Hubbard projectors, we found considerable deviations in the band gaps, with the orthogonalized projectors showing the highest accuracy, in overall close agreement with experimental data.

Additionally, we have analyzed systematically which electronic states contribute predominantly to the valence band maximum and to the conduction band minimum in order to determine the states that might require Hubbard corrections. We have found that in the 14 TM-containing materials the *d* states of TM elements are mostly found at the bottom of conduction bands while the *p* states of light elements (O, N, S) appear at the top of the valence bands. We have not found any clear trend as to whether one should apply the *U* correction to both groups of states or only to the *d* states of TM elements—either of these two options perform equally (for the band gaps) or one of them is better than the other depending on the specific material. We note that all materials were considered to be nonmagnetic in our simulations, and hence future studies with more detailed investigation of various magnetic orderings would be useful. As for the remaining 6 materials containing the *p*-block (group III-IV) elements, we found that the *p* states of light elements (O, N, S) contribute mainly to the top of the valence bands and the *p* states of the group III-IV elements appear mostly at the bottom of the conduction bands. Application of the *U* corrections to both groups of states or only to the *p* states of light elements give essentially the same band gaps, which means that the application of *U* corrections to the *p* states of group III-IV elements leads to minor improvements (if any).

Hence, this study conclusively shows that DFT+*U* with *ab initio U* may serve as a consistent and reliable electronic-structure method for improving band gaps beyond (semi)local functionals, provided that care is taken in choosing appropriate Hubbard projectors.

**Author Contributions:** Author contributions include conceptualization, I.D.; methodology, I.T.; software, Y.X., N.E.K.-H., W.Z., and I.T.; validation, N.E.K.-H. and W.Z.; formal analysis, N.E.K.-H.; investigation, N.E.K.-H. and W.Z.; resources, I.D.; data curation, N.E.K.-H. and W.Z.; writing–original draft preparation, N.E.K.-H.; writing–review and editing, I.D., I.T., W.Z., and Y.X.; visualization, N.E.K.-H.; supervision, I.D. and I.T.; project administration, I.D.; funding acquisition, I.D. All authors have read and agreed to the published version of the manuscript.

**Funding:** This research was funded by the DMREF and INFEWS programs of the National Science Foundation under Grant Agreement No. DMREF-1729338, and by the Swiss National Science Foundation (SNSF) through grant 200021-179138 and its National Centre of Competence in Research (NCCR) MARVEL.

**Institutional Review Board Statement:** Not applicable.

**Informed Consent Statement:** Not applicable.

**Data Availability Statement:** The data used to produce the results of this work are available in the *Materials Cloud Archive* [137] and through the *HydroGEN DataHub* website under the project NSF DMREF PSU PEC.

**Conflicts of Interest:** The authors declare no conflict of interest.

## Appendix A. Convergence Tests for the Self-Consistent Calculation of Hubbard Parameters

In this Appendix we discuss how we performed convergence tests when computing Hubbard parameters using the DFPT approach (see Section 2.1). We did these tests for $TiO_2$, NbNO, and $Y_2SO_2$ which represent the material families considered in this study. As explained in Ref. [79], Hubbard $U$ must be converged with respect to the **k**- and **q**-points sampling of the BZ: the former is the Bloch wavevector that is used to describe Kohn-Sham wavefunctions and charge density, while the latter is used to describe modulations of monochromatic perturbations that are used in linear-response calculations when computing $U$ (see Section 2.1). As the criterion to monitor the convergence of $U$ we use the ratio $N_{\mathbf{k}}/N_{\mathbf{q}}$, where $N_{\mathbf{k}}$ and $N_{\mathbf{q}}$ are the number of **k**- and **q**-points in the BZ, respectively. It is important to remark that we use this criterion with the condition that the **k**-point mesh for each material remains unchanged (i.e., it is fixed such that the spacing between the **k**-points is 0.04 Å$^{-1}$). The result is shown in Figure A1.

In these plots, convergence occurs from right to left, i.e., the smaller the ratio $N_{\mathbf{k}}/N_{\mathbf{q}}$ the denser the **q**-point mesh at a fixed **k**-point mesh. For example, $N_{\mathbf{k}}/N_{\mathbf{q}}=1$ is the case where the **q**-points are equivalent to the **k**-points for a given material and hence, the most computationally expensive option (in this case the spacing between the **q**-points is also 0.04 Å$^{-1}$). Since we enforce a consistent **k**-point sampling density in our calculations, **k**-points are unique for each material in this study. Thus, when we choose the number of **q**-points for the sampling of the BZ corresponding to the primitive cell, we look at the $N_{\mathbf{k}}/N_{\mathbf{q}}$ ratio by which we divide the **k**-points for each material and truncate down to the nearest integer value. For example, if the **k**-point mesh is $6 \times 5 \times 5$ and we have a $N_{\mathbf{k}}/N_{\mathbf{q}}$ ratio of 2, then the resulting **q**-point mesh is $3 \times 2 \times 2$. The $N_{\mathbf{k}}/N_{\mathbf{q}}$ ratios tested, as well as their subsequent **q** meshes for each material are shown in Table A1. From the **q**-point mesh testing, a $N_{\mathbf{k}}/N_{\mathbf{q}}$ ratio of 2 was selected with a convergence threshold of $\leq 0.1$ eV. By selecting a higher $N_{\mathbf{k}}/N_{\mathbf{q}}$ ratio than 1, we are able to speed up Hubbard parameter calculations and reduce the computational cost.

Once the convergence of $U$ is reached with respect to the **q**-point sampling of the BZ at a fixed **k**-point sampling, the next step is to perform the self-consistency convergence test. This is explained in detail in Ref. [38]. In short, it is necessary to reach self-consistency between the computed value of Hubbard $U$ and the crystal structure; to do so, we need to perform structural optimization at the DFT+$U$ level of theory with the current value of $U$, then recompute $U$ using DFPT on top of the new geometry, then perform new structural optimization with the updated value of $U$, and so on until convergence is reached [38]. The results for the selected systems are shown in Figure A2. It can be seen that the accuracy of $\Delta U \leq 0.1$ eV is reached after 3 cycles for all systems.

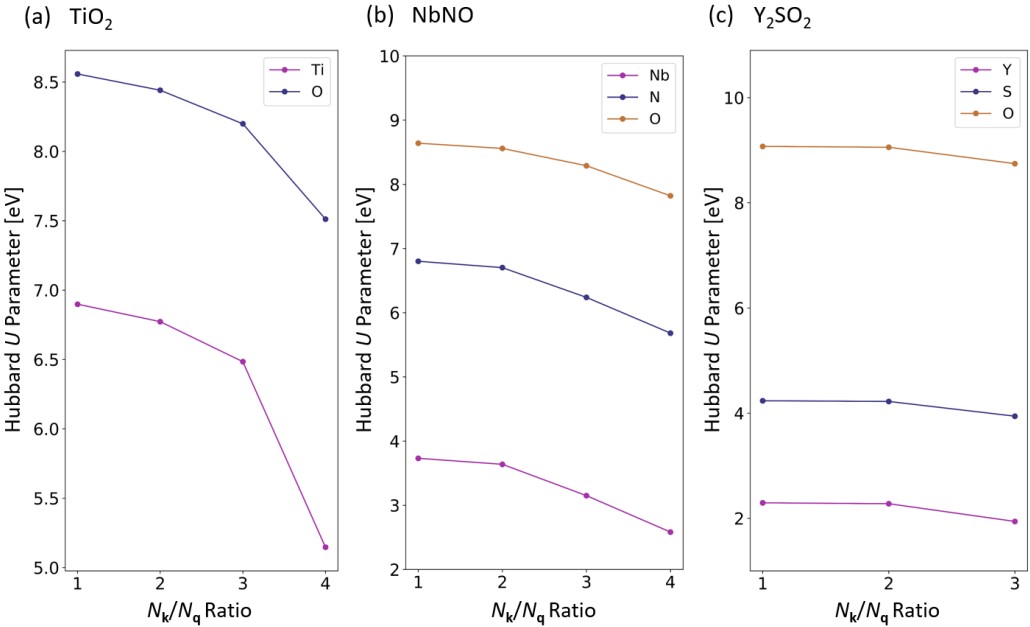

**Figure A1.** Dependence of the Hubbard $U$ parameter on the $N_{\mathbf{k}}/N_{\mathbf{q}}$ ratio showing the results of $N_{\mathbf{k}}/N_{\mathbf{q}}$ testing for 3 representative materials: (**a**) $TiO_2$, (**b**) NbNO, and (**c**) $Y_2SO_2$. For each plot, the **q** points density increases from right to left, where at $N_{\mathbf{k}}/N_{\mathbf{q}}$=1, the **q**-point mesh matches the **k**-point mesh. In panel (**c**), $Y_2SO_2$ does not have a $N_{\mathbf{k}}/N_{\mathbf{q}}$ ratio of 4 plotted because its $N_{\mathbf{k}}/N_{\mathbf{q}}$ ratio of 3 results in a **q** mesh equivalent to that produced using a $N_{\mathbf{k}}/N_{\mathbf{q}}$ ratio of 4 (see Table A1).

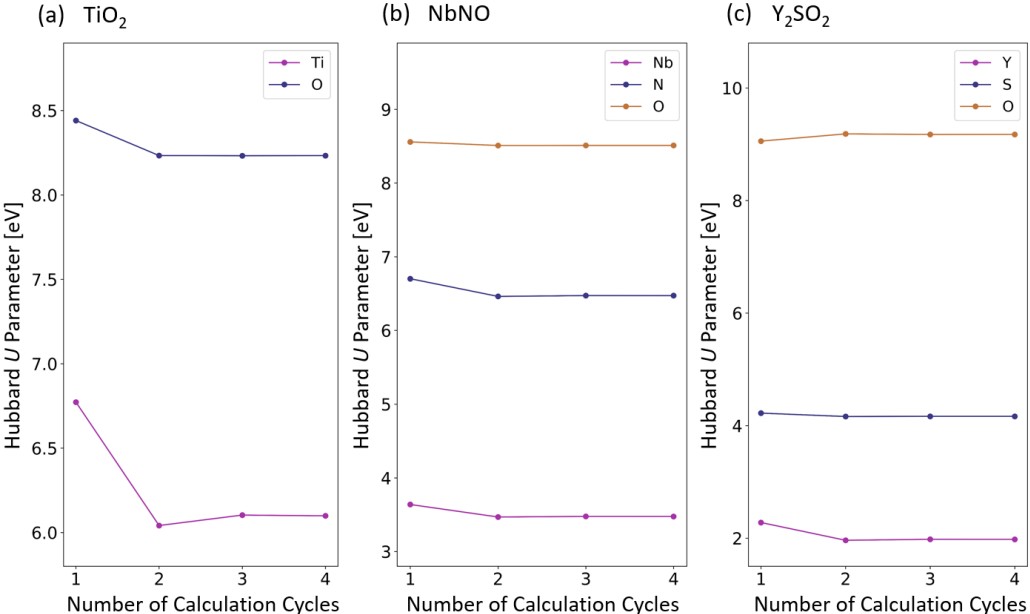

**Figure A2.** Convergence of the Hubbard $U$ parameter with respect to the number of self-consistent cycles for 3 representative materials from our dataset: (**a**) $TiO_2$, (**b**) NbNO, and (**c**) $Y_2SO_2$.

**Table A1.** Comparison of $\mathbf{q}$-point meshes for each material ($TiO_2$, NbNO, $Y_2SO_2$), corresponding to $N_{\mathbf{k}}/N_{\mathbf{q}}$ ratios of 1, 2, 3, and 4. For the case where $N_{\mathbf{k}}/N_{\mathbf{q}}=1$, the $\mathbf{q}$-point mesh is equivalent to the $\mathbf{k}$-point mesh.

| Formula | k-Point Mesh | $N_{\mathbf{k}}/N_{\mathbf{q}}$ Ratio | | | |
|---|---|---|---|---|---|
| | | 1 | 2 | 3 | 4 |
| $TiO_2$ | $9 \times 6 \times 6$ | $9 \times 6 \times 6$ | $4 \times 3 \times 3$ | $3 \times 2 \times 2$ | $2 \times 1 \times 1$ |
| NbNO | $6 \times 5 \times 5$ | $6 \times 5 \times 5$ | $3 \times 2 \times 2$ | $2 \times 1 \times 1$ | $1 \times 1 \times 1$ |
| $Y_2SO_2$ | $8 \times 8 \times 4$ | $8 \times 8 \times 4$ | $4 \times 4 \times 2$ | $2 \times 2 \times 1$ | $2 \times 2 \times 1$ |

## Appendix B. Computational Cost Comparison of HSE06 and Hubbard Parameter Calculations

In this Appendix we present a comparison of the computational costs of hybrid functional (HSE06 [96,97]) calculations and linear-response calculations of Hubbard $U$ parameters [138]. This comparison is made for two representative materials, $TiO_2$ and NbNO. These materials were selected as examples of relatively small unit cells (rutile $TiO_2$ has 6 atoms per unit cell) and relatively large unit cells (NbNO has 12 atoms per unit cell)—in comparison to the diverse materials studied in this work. HSE06 self-consistent-field calculations were performed using the optimized geometry of the materials at the DFT+$U$ level of theory. The computational setup is the same as described in Section 3. We recall that hybrid functional calculations are computationally very expensive because the calculation of the non-local exact-exchange potential contains double sums over electronic bands and double sums over $\mathbf{k}$-points (one of these $\mathbf{k}$-point meshes is replaced by a scarcer $\mathbf{q}$-point mesh to speed-up the calculations). As said above, it is common to lower the computational cost of HSE06 calculations (with some losses in the accuracy) by reducing the number of $\mathbf{q}$-points relative to the $\mathbf{k}$-points of a given material by setting a ratio $N_{\mathbf{k}}/N_{\mathbf{q}}$ higher than 1 [139]. For the current HSE06 calculations, $N_{\mathbf{k}}/N_{\mathbf{q}}$ was set to 3 for each material [140]. The resulting $\mathbf{k}$- and $\mathbf{q}$-point meshes for the $U$ linear-response and HSE06 calculations are listed in Table A2 (we stress that the $\mathbf{q}$-point sampling has different physical meaning in these two types of calculations). The Hubbard parameters calculations reported in Table A2 were performed using a $N_{\mathbf{k}}/N_{\mathbf{q}}$ ratio of 2, as discussed in Appendix A.

**Table A2.** Comparison of the computational costs of the one-shot $U$ linear-response and HSE06 calculations for $TiO_2$ and NbNO. The ratio between the time for HSE06 ($t_{\text{HSE06}}$) and $U$ ($t_U$) calculations is given by $t_{\text{HSE06}}/t_U$, and the band gaps $E_g$ calculated using HSE06 are also reported (in eV).

| Formula | No. of Atoms per Unit Cell | k-Points Mesh | $U$ Calculation | | HSE06 Calculation | | | Ratio |
|---|---|---|---|---|---|---|---|---|
| | | | q-Points | $t_U$ | q-Points | $E_g$ | $t_{\text{HSE06}}$ | $t_{\text{HSE06}}/t_U$ |
| $TiO_2$ | 6 | $9 \times 6 \times 6$ | $4 \times 3 \times 3$ | 3h 46m | $3 \times 2 \times 2$ | 3.30 | 10h 1m | 2.7 |
| NbNO | 12 | $6 \times 6 \times 6$ | $3 \times 2 \times 2$ | 7h 35m | $2 \times 2 \times 2$ | 2.68 | 61h 52m | 8.2 |

It is clear from Table A2 that the HSE06 calculations are computationally more expensive than Hubbard $U$ calculations, even in this particular case where a lower sampling density of $\mathbf{q}$-points was used for HSE06 ($N_{\mathbf{k}}/N_{\mathbf{q}} = 3$) than for the Hubbard $U$ calculations ($N_{\mathbf{k}}/N_{\mathbf{q}} = 2$) (even though these two $\mathbf{q}$-points samplings are not really directly comparable). Additionally, it is seen from Table A2 that the relative computational cost increases rapidly with the system size, with the HSE06 calculations being 2.7 times more costly than Hubbard $U$ calculations for $TiO_2$ and 8.2 times more computationally costly than Hubbard $U$ calculations for NbNO. However, it is important to remark that in Table A2 we report the timing of only the one-shot calculation of the $U$ parameters, while in practice we need to use the self-consistency procedure that requires typically 2-3 such calculations (see Appendix A). Thus, by considering the factor of 3 that was used in this work for the self-consistency loop, we obtain the ratios $t_{\text{HSE06}}/t_U$ of 0.9 for $TiO_2$ and 2.7 for NbNO. Hence, we can still see that DFT+$U$ with self-consistent $U$ values is more advantageous for systems with more than ~10 atoms in the unit cell. In fact, in Ref. [36] it was shown that DFT+$U$ with $U$ computed

using linear response theory can be used for systems as large as 320 atoms in the supercell, while this is beyond the reach for hybrid functionals with the current high-performance computing hardware. However, we note that the comparison shown in Table A2 is very approximate as each of these calculations can be optimized and speed-up further (e.g., by lowering the kinetic-energy cutoff for the density and potentials for the exact-exchange term in the HSE06 calculations, by lowering the convergence threshold for the response matrices in the $U$ calculations (see Equation (6)), by using more efficient parallelization strategies, for instance).

Moreover, it is important to note that not only the computational cost matters when comparing HSE06 with DFT+$U$ calculations, but also the accuracy of the final results. In fact, we find that the band gap value predicted for $TiO_2$ using HSE06 (see Table A2) is in slightly better agreement with the experimental value of 3.0 eV than the best band gap value predicted using DFT+$U$ of 2.60 eV (see Table 3), however the band gap value predicted for NbNO using HSE06 is in much worse agreement with the experimental value of 2.1 eV than the best band gap value predicted using DFT+$U$ of 2.05 eV (see Table 3). This observation can be explained by the fact that in the fully first-principles DFT+$U$ approach, $U$ is a material-specific parameter and it is computed *ab initio* as a response property of the material. In contrast, in HSE06 (and other popular hybrids) the amount of the exact-exchange (the screening parameter) is fixed (i.e., it is not material-specific) and in solids this often leads to unsatisfactory results. Thus, frequently this screening parameter is tuned until the experimental value of some property (e.g., the band gap) is reproduced, which makes this approach not fully first-principles. However, it is important to mention that this screening parameter in hybrids can be computed *ab initio* (see, e.g., Refs. [13,15,18]), which is not always an easy task. In this case, an extra computational cost for HSE06 must be added in Table A2 which will make the ratio $t_{HSE06}/t_U$ even larger.

Therefore, the overall observations in this appendix further confirm that DFT+$U$ can be considered as a reliable and efficient method to calculate band gaps at a fraction of the computational cost of hybrids functionals such as HSE06 (especially for large systems).

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
