# Peer review of "Extensive Benchmarking of DFT+U Calculations for Predicting Band Gaps"

_applsci, doi:10.3390/app11052395_

Round 1

Reviewer 1 Report

Authors present a study that shows that DFT+U (with ab initio U) with very carefully chosen Hubbard projectors may serve as a consistent and reliable electronic-structure method for improving band gaps beyond (semi)local functionals.

The paper is very clearly written and organized.

Minor problem is that the figures have relatively small lettering which makes the partly hard to digest. Maybe try to increase the font at the palces where it is possible.

Reviewer 2 Report

This is a very well written paper on a subject matter of considerable interest. The goal of the paper is to develop accurate theoretical approach(es) to determine optical gaps of inorganic semiconductors (mostly binary). The authors have done an excellent job of presenting the nature and scope of the problem, and presenting their results in Tables 2-4 and Fig. 2. The authors make a strong case for the DFT+U approach to calculating band gaps. Importantly, the authors are also careful in staying out of similar calculations for organic semiconductors, where the exciton binding energy is very large. 

I recommend publication as is.

Reviewer 3 Report

Dear uthors,

Your manuscript targets a hot topic and the results can be relevant to the scientific community and the journal readers. Unfortunately some major and minor issues have to be fixed before I can suggest acceptance of your manuscript.

Major isuue:

you claim in your manuscript that "This work demonstrates that DFT+U may serve as a useful method for high-throughput workflows that require reliable band gap predictions at low computational cost." here I just would like you to provide an example of such large screening. Why such data that validate this assumption is not provided in the manuscript.

Minor issue:

Please provide a comparaison of the different methods for prediction of band gaps including costs and time required to provide the reader with a synthetic view of the field.

Without adressing these issues the manuscript could not provide valuable informations to the readers of the journal

Best regards
